# Effects of Data Augmentations on Speech Emotion Recognition

**DOI:** 10.3390/s22165941

**Published:** 2022-08-09

**Authors:** Bagus Tris Atmaja, Akira Sasou

**Affiliations:** 1National Institute of Advanced Industrial Science and Technology, Tsukuba 305-8560, Japan; 2Institut Teknologi Sepuluh Nopember, Surabaya 60111, Indonesia

**Keywords:** speech emotion recognition, affective computing, data augmentations, wav2vec 2.0, SVM

## Abstract

Data augmentation techniques have recently gained more adoption in speech processing, including speech emotion recognition. Although more data tend to be more effective, there may be a trade-off in which more data will not provide a better model. This paper reports experiments on investigating the effects of data augmentation in speech emotion recognition. The investigation aims at finding the most useful type of data augmentation and the number of data augmentations for speech emotion recognition in various conditions. The experiments are conducted on the Japanese Twitter-based emotional speech and IEMOCAP datasets. The results show that for speaker-independent data, two data augmentations with glottal source extraction and silence removal exhibited the best performance among others, even with more data augmentation techniques. For the text-independent data (including speaker and text-independent), more data augmentations tend to improve speech emotion recognition performances. The results highlight the trade-off between the number of data augmentations and the performance of speech emotion recognition showing the necessity to choose a proper data augmentation technique for a specific condition.

## 1. Introduction

The development of speech emotion recognition (SER) has been directed in either finding the new features correlated to emotion or building classifiers suited for emotion recognition problems. The initial survey on SER addressed features and classification schemes as the most important aspects of designing SER besides the database or dataset [1]. A recent survey on SER [2] also has shown that features (including pre-processing step) and classifiers are the most studied areas among others in SER.

Instead of proposing a new method, investigating the impact of data augmentation is worth studying. For example, reference [3] found that building self-supervised learning automatic speech recognition results in extremely large and diverse datasets. However, for speech emotion recognition, the availability of the dataset is not as large as speech recognition datasets. Therefore, research on data augmentation is a good opportunity to investigate the impact of data augmentation on SER performance.

Data augmentation has been found useful in areas outside speech emotion recognition. Ko et al. [4] evaluated a low-cost data augmentation technique by speed perturbation and found that it improved the word error rate (WER) over other methods. Casanova et al. [5] employed both transfer learning and data augmentation for improving COVID-19 detection from cough sounds. Similarly, data augmentation could improve SER performance in specific ways. However, no study has been found regarding these impacts of data augmentation on SER performance.

In view of the above assumption based on findings in the other research areas, this paper proposes a systematic investigation of the effect of data augmentation on speech emotion recognition. The authors find no systematic investigation has been conducted on the effect of data augmentation on speech emotion recognition. This paper concretely contributes mainly to the following investigations:the data augmentation types that contribute to the performance of speech emotion recognition;the number of data augmentations that optimally improves the performance of speech emotion recognition.

The authors have experimented with two datasets and four data augmentation techniques. The datasets are Japanese Twitter-based emotional speech (speaker-independent, text-independent, and speaker+text-independent) and IEMOCAP (speaker-independent). Four types of data augmentation include glottal source extraction, silence removal, impulse response convolution, and noise addition. The number of data augmentations is the number combination of four types of data augmentation above: one augmentation, two augmentations, three augmentations, and four augmentations. Only augmentations based on the raw audio signal are considered in this paper. Other augmentation types (e.g., spectrogram-based augmentation) are not adopted due to the difficulties of mixing these types of data augmentation with raw audio signals.

## 2. Previous Work

Research on speech processing has been focused on developing new methods instead of evaluating existing methods with different evaluation conditions. Nevertheless, evaluating the training conditions for such speech processing goals is a challenging task in addition to developing the new methods.

In automatic speech recognition (ASR) research, which is the main task of speech proceedings, an evaluation of audio augmentation for ASR has been conducted to observe the effect of different speeds of the audio signal on ASR performance [4]. Using different speed factor of 0.9, 1.0, and 1.1, which is low-cost and easy to adopt, the authors found that the WER of ASR performance is improved by 4.3% in four tasks. Similar audio augmentation techniques may also work to improve the performance of SER.

In [6], the authors evaluated training data selection between neutral and emotional speech for speech recognition. The results showed that training a sufficient amount of spontaneous data was more beneficial than a small amount of emotional speech. In other words, emotional speech is harder to be recognized than neutral speech for speech recognition. Horii et al. [7] analyzed the feature extraction method by convolution neural network and obtained an accuracy of 62% on the JTES dataset.

Research on SER has been conducted actively on the English dataset. IEMOCAP [8] is the most widely used dataset for speech emotion recognition. Evaluations on general speech processing tasks, including SER, have been performed to evaluate the effectiveness of self-supervised learning methods [9]. The evaluation showed the superiority of HuBERT model on most tasks, including SER. HuBERT achieved a weighted accuracy of 67.72% on the IEMOCAP dataset with five-fold cross-validation. Fusing acoustic with linguistic features (from transcription) improved the overall accuracy to 77.51% on cross-validation evaluation and 83.08% on Session 5 evaluation [10]. Adapting speaker awareness to build a pre-trained model for universal speech representations [11] obtained an overall accuracy of 70.78% on the IEMOCAP cross-validation evaluation.

Research on Japanese SER has been actively developed since the construction of JTES dataset [12]. Using this dataset, Lee [13] achieved an accuracy of 81.61% for speaker-independent evaluation with deep neural network architecture. Nagase et al. [14] obtained UA of 71.31% with acoustic features (speaker-independent), 66.34% with linguistic features, and 91.22% with speaker+linguistic features (text-independent). Research on this Japanese SER has been continued by evaluating different splitting criteria, feature extraction methods, and classifiers.

In [15], the authors investigated the splitting criteria for training speech emotion recognition on the JTES dataset, whether splitting the training data by speakers (speaker-independent), by texts (text-independent), or both (speaket+text-independent). The results showed that splitting by speakers and text was the most difficult among others. It is difficult for the SER model to infer new data with different linguistic information from training data. The current model needs to be trained on the same linguistic information to obtain sufficient performance.

In [16], the authors evaluated bidirectional LSTM methods based on multi-stream attention with feature segmentation on JTES dataset. The authors highlighted that the key to their SER performance improvements was data augmentation (max. UA for JTES speaker+text-independent is 73.4%). However, no direct comparison of the same method with and without data augmentation was found. This gap is left; however, it is important to study the contribution of data augmentation on SER to direct future work.

This study aims to fill that gap. We wanted to compare the performance of emotion recognition among different data augmentation types and the number of data augmentations. As an additional analysis, we perform a benchmark on the JTES and IEMOCAP datasets, which can be used as a comparison tool for the current study and future studies.

## 3. Methods

### 3.1. Dataset

#### JTES

Japanese Twitter-based emotional speech (JTES) corpus [12] is evaluated in this research. The dataset contains 20,000 utterances from 100 speakers, four emotion categories, and 50 sentences. The original sampling rate was 48 kHz but was resampled to 16 kHz during the feature extraction process. From all samples in the dataset, we choose speaker-independent (SI), text-independent (TI), and speaker and text-independent (STI) criteria following the previous research [15,16]. The SI and TI criteria result in 16,000 utterances for training and the rest 4000 utterances for the test (80/20 split). The STI criterion results in 14,400 samples for training and 400 samples for the test. We excluded the speaker-dependent (SD) criterion from the previous study [15] since SI is more challenging than SD [17] and is a standard evaluation in most speech processing tasks, including SER. For training data, we performed data augmentations in addition to the original data. For the test data, we did not add any data augmentation technique. The test set is kept as it is to enable benchmarking with previous methods. Details of the dataset, including its visualization, can be found in the previous studies [12,15]. Figure 1 shows the flow of training data selection to investigate the data augmentation on SER.

#### IEMOCAP

The interactive emotional dyadic motion capture (IEMOCAP) database is one of the most common datasets for speech emotion recognition tasks. The dataset focuses on understanding expressive human communication through a combination of verbal and non-verbal channels from both speech and gestures. The original corpus contains 10,039 utterances/sentences with a 16 kHz sampling rate; this research utilized a subset of that total utterance commonly evaluated for the four categories of emotion [18,19,20]. The four emotion categories are neutral, happy, anger, and sadness. The excitement emotion category is merged with the happy emotion. Details of dataset, including the visualization and distribution, can be found in the previous studies [8,20,21,22]. From five sessions in the dataset, Sessions 1–4 are allocated for training, while Session 5 is for a test. This leave-one-session-out (LOSO) evaluation is speaker-independent since the speakers for each session are different (two speakers per session). As in JTES, the augmentations are performed for training but not for the test set.

### 3.2. Data Augmentations

Four data augmentations are applied to the original datasets. The choice of different data augmentations is to determine the type of data augmentation that contributes to the training of speech emotion recognition. Furthermore, these four data augmentations can be combined to evaluate the optimal number of data augmentations for SER in this study. The data augmentation techniques are described in the following sections.

#### 3.2.1. Glottal Inverse Filtering (Glottal Source Extraction, glt)

Several studies [23,24,25,26,27] have investigated the characteristics of glottal flow regarding different emotional states and have indicated that using glottal inverse filtering (GIF)-based speech analysis, emotional cues can be obtained from speech signals. In this data augmentation, we adopt a GIF method based on a constrained auto-regressive hidden Markov model (CAR-HMM) [28]. The CAR-HMM extracts the glottal flow derivative from a voiced speech by introducing some gain constraints to the AR filter. The glottal flow derivative is defined by a glottal flow that is combined with lip radiation characteristics. The gain constraints to the AR filter are introduced to avoid some characteristics of the glottal flow derivative remaining in the estimated AR filter.

The CAR-HMM, as depicted in Figure 2, is a kind of source-filter model, where the AR filter and the HMM, as depicted in (b) and (f), represent a vocal tract and a generative model of a glottal flow derivative, respectively. The states in the HMM are concatenated in a ring form so that the state transition circulates to represent the periodicity of the voiced speech. Each state has an output probability distribution (PDF) of a single Gaussian distribution defined by an expectation and a variance. Given a state transition sequence, the expectations and variances of the output PDFs can align (for example) as depicted in (d) and (e), respectively. The glottal flow derivative, as depicted in (c), is then defined by the realized values of the aligned output PDFs. Finally, the CAR-HMM is assumed to generate the voiced speech as depicted in (a) by filtering the glottal flow derivative with the AR filter.

In the CAR-HMM-based speech analysis, the AR coefficients, the glottal flow derivatives, and the corresponding HMM parameters must be estimated using a given voiced speech. The iteratively-parameter estimation algorithm of the CAR-HMM is adopted to obtain these values. The details have been described in [6]. In this augmentation, the prediction order of the AR filter and the number of HMM states are set to be 16 and 20, respectively. The AR filter’s gains at both the DC and the Nyquist frequency are constrained to be 1.

#### 3.2.2. Silence Removal (Speech Cleaned, spc)

In this data augmentation, we remove silence durations before the beginning point and after the endpoint of each utterance (start-end silence removal).

#### 3.2.3. Applying Impulse Response (ir)

In this data augmentation, we convolve original JTES audio data with a random impulse response. Two impulse response datasets are evaluated [29,30]. The amplitude of the impulse response is set to be 0.5 from the maximum amplitude of the corresponding audio signal.

#### 3.2.4. Noise Addition (noi)

In noise addition data augmentation, we mix original audio files in the JTES dataset with noises from the Environmental Sound Classification dataset with 50 classes (ESC-50) [31]. The ESC-50 consists of 2000 noises in five major categories: animal, natural, human, interior, and exterior sounds. The augmentation method randomly chooses one noise file and mix it with the original audio file.

Figure 3 shows the example of data augmentations in the JTES dataset with data augmentation techniques above. The most left is the original sample from the dataset. The rest are augmented data by glottal source extraction (spc), silence removal (spc), applying impulse response (ir), and adding noise (noi). It can be seen clearly the difference among data. For instance, in glt and spc, the duration of the signal becomes shorter than in others, while in ir and noi the shape of the signals is changed due to impulse response convolution and noise addition.

In addition to the four types of data augmentation techniques, we also evaluated the performance of combinations of the four data augmentations above. The results of the combinations are six arrangements for two data augmentations, four arrangements for three augmentations, and one arrangement for four data augmentations. The number of training data on each data augmentation is shown in Table 1.

### 3.3. Acoustic Features

We employed a model for dimensional speech emotion recognition [32,33] based on wav2vec 2.0 [34] to extract speech embeddings as acoustic features. This model is trained on MSP-IMPROV dataset [35]. The size of the embedding is a vector of 1024-dimensional, which is the number of hidden states in the model. The pre-trained model extracts speech embeddings on both JTES and IEMOCAP datasets with the same configuration.

### 3.4. Classifier

A support vector machine (SVM) for classification (SVC) is chosen as the classifier. SVM is one of the most common classifiers for SER tasks [36]. Other SER research employed MLP [17], LSTM [16,37,38], or combinations of CNN with LSTM [19,39,40]. This study chose SVM since it is the simplest classifier among them. This study focuses on investigating the effect of different data augmentations instead of proposing a new classifier architecture. The acoustic feature described in the previous subsection is standardized by removing the mean and scaling to unit variance before it is fed into SVC. The SVC is implemented through the scikit-learn library [41]. We left the configuration of SVC as default except for the kernel coefficient. The kernel coefficient is set to 1/1024 (`auto’). The same configuration of SVC classifier applies to both JTES and IEMOCAP datasets.

### 3.5. Evaluation Metric

We evaluated the performances of data augmentations on SER with a single metric, namely unweighted average recall (UAR). This metric is an average recall of predictions over the true labels for each emotion category. This metric is also known as balanced accuracy or unweighted accuracy (UA).

The methodology, from the datasets to the evaluation metric, is shown in Figure 4. The methods, excluding the datasets, are open to the public in the following repository: https://github.com/bagustris/ser_aug (accessed on 12 July 2022).

## 4. Experiments

We conducted two experiments on the different data augmentations and apparatus (computing systems) to generalize the performances of evaluated systems on different machines. The main aim of using two apparatus is to investigate the effect of adding random impulse responses and noises to the original data aside from checking the consistency of results for adding glottal source filtering and removing start-end silences.

### 4.1. Experiment 1 (Exp. #1)

In experiment 1, we evaluated all data augmentation techniques described previously on a PC with Ubuntu 18.04 OS, AMD EPYC 7702P 64-Core CPU, 504 GB RAM, and two NVIDIA GeForce RTX 3090 cards (24 GB/card). For impulse response addition, we used audio files from the MIT Acoustical Reverberation Scene Statistics Survey [29]. For noise addition, we used noise files from ESC-50: Dataset for Environmental Sound Classification [31].

### 4.2. Experiment 2 (Exp. #2)

In experiment 2, we evaluated all data augmentation techniques similarly to Experiment 1 on an HPC (abci.ai) with CentOS Linux 7, Intel Xeon Gold 6148 CPU, 360 GB RAM, and NVIDIA Tesla V100 (16 GB). For impulse response addition, we used audio files from the EchoThief Impulse Response Library [30]. For noise addition, we used the same noise files from ESC-50: Dataset for Environmental Sound Classification [31]. Note although the noise dataset is the same, the results of data augmentation are different since the audiomentation package version 0.19.0 [42], which was used to mix original data with noise, chose the noise files randomly to mix in.

## 5. Results and Discussion

Table 2, Table 3, Table 4 and Table 5 show the results of experiments. The results of two investigations—the type and number of augmentations—can be inferred from each table. The results with one augmentation depict the investigation of the type of data augmentation that contributes to SER performances. The rest of the results depict the investigation of the effect of the number of data augmentations. We measured those performances of the data augmentation techniques on SER with the UA/UAR metric on 0–100% scale. These results on two experiments are deterministic, meaning that the same scores are obtained for the same data for different runs on the same apparatus.

### 5.1. JTES-SI

The first evaluation is the speaker-independent criterion, a standard criterion in most speech processing tasks. The result shown in Table 2 exhibits higher performances than the previous study [13]. Although the author has proposed a method for multilingual SER in that paper, comparing monolingual SER with the same JTES dataset between that study and this study shows performance improvement, even for the results without augmentation in this study. The key ingredient of our study is the utilization of deep learning-based acoustic feature extraction, which used a pre-trained model from a specific emotional speech database using wav2vec 2.0 rather than neutral speech. The previous study [13] used INTERSPEECH 2010 acoustic feature set, which is non-deep learning-based feature extraction.

For this SI evaluation, the best improvement has been achieved by the largest number of data augmentations, i.e., four data augmentations with glt, spc, ir, and noi. The improvement is about 1% from the baseline data without augmentation. The individual augmentation (with one augmentation) shows degradation (orig + glt), and improvements with orig + ir obtained the highest score for an additional type of augmentation. The trend in this SI evaluation yields a presumption that more augmentation will improve SER performance.

### 5.2. JTES-TI

The text-independent evaluation is a new splitting criterion for SER [15,43]. Those previous studies have shown that text-independent evaluation is more difficult and challenging than traditional speaker-independent evaluation, and this study supports that finding. Compared to SI with 97.10% UA for baseline, TI obtained 75.08% of UA. In contrast to a previous study that evaluated the same number of test set for comparing SI, TI, and STI [15], this study evaluated a different number of test set between SI and TI to take advantage of the number of samples/utterances. By evaluating 2000 samples as a test set, this study revealed a similar pattern to the previous study on the same JTES dataset.

Different portions of the training/test split were also evaluated in this study (16,000/4000) compared to the previous study [15] (19,600/400 and 14,400/400). However, the trends look similar in both studies. Both SI and TI improved about 10% from that previous study. The key factor, as explained above, is the different acoustic features used in this study with a less sophisticated classifier (SVM in this study vs. MLP in the previous study).

The best data augmentation in this TI evaluation is obtained by glottal source filtering. This approach leads to 1.35% of UA improvement. These results are opposite to the previous JTES-SI evaluation in which glt obtained the worst score among other types of augmentation. In this TI evaluation, more data augmentation does not tend to increase the performance of SER. By using four data augmentations, the results are only improved by about 0.5% from the baseline. The highest UA for this TI evaluation is obtained by augmenting the original data with glt and spc (two augmentations), which yields the results of 76.75% UA.

### 5.3. JTE-STI

For JTES-STI evaluations, we found that cleaning original speech signals by removing silences is the most effective data augmentation technique by 2% UAR improvement. The other two data augmentations, the glottal source extraction and noise addition, slightly improved SER’s performance. Augmenting SER with the impulse response in experiment 2 decreases the performance; the reverberation by room shapes degrades the recognition rate of emotional speech.

For the number of data augmentations in STI, we found that the number of data augmentations with two augmentations is the most effective data augmentation technique by 3% UAR improvement from the baseline without augmentation. Among six arrangements of two data augmentations, augmented original JTES dataset with glottal source extraction and silence removal consistently achieved the highest UAR on both experiment 1 and experiment 2. These results are also in line with the types of data augmentation techniques (with one augmentation), in which data augmentation with silence removal (speech cleaned, spc) topped the performance among other data augmentation types.

Compared to the previous work [15], we observed the different pattern for JTES-SI, JTES-TI, and JTES-STI. In this work, the performance of JTES-STI and JTES-TI on original data without augmentation is similar with JTES-TI (UAR = 75.08%), slightly higher than JTES-STI (UAR = 74.5%). With augmentation, the results are inverted. JTES-STI (UAR = 77.25%) obtained slightly higher results than JTES-TI (UAR = 77.75%). The small set of JTES-STI test data (400 samples) may affect the performance results. This study evaluated 2000 samples of JTES-TI and 400 samples of JTES-STI for the test (see Table 1 for details). Hence, it is not fair enough to compare JTES-STI directly and JTES TI here. There may be no significant differences between text-independent and speaker+text-independent evaluations; however, further study is needed to clarify this assumption.

### 5.4. IEMOCAP

IEMOCAP with speaker-independent evaluation leads to the results shown in Table 5. As in the JTES evaluations, the improvement from the original data without augmentation to four data augmentations is about 1.5% of UA. In this IEMOCAP-SI evaluation, more data augmentations tend to improve SER performance as in JTES-SI evaluation.

By comparing individual data augmentations, it is not clear which type of data augmentations performs better than others. Augmenting original data with start-end silence removal and noise addition yields similar results to each other in experiment #1. These results are consistent (no one type of data augmentation is superior to the others) with previous results on JTES dataset evaluations in this study.

The highest result with four augmentations is achieved in experiment #1. The result of experiment #2 for this number of data augmentation is only lower than that of three data augmentations with spc + ir + noi (gap of 0.07%). By inspecting this number, we can presume that more data augmentations tend to yield performance improvement. In this study, this trend is observed for both JTES and IEMOCAP datasets with speaker-independent evaluations.

### 5.5. Comparison to the Previous Studies

As an additional analysis, we performed a literature study conducted similarly on the JTES dataset with a similar split scenario. We found that our results are competitive with the results of the literature studies for both with and without data augmentation. Table 6 shows the benchmark of our results with the literature studies. It is also clear from Table 6 that our acoustic feature engineering contributes more than data augmentation techniques. Without data augmentation, wav2vec 2.0 pre-trained on MSP-IMPROV dataset [35] achieved higher UAR than other proposed methods. These results highlight the importance of using pre-trained model wav2vec 2.0 trained on affective dataset over other techniques. Note that reference [14] only reported speaker-independent only for their acoustic-only SER while reference [39] reported text-dependent SER on the JTES dataset. A split scenario with speaker+text-independent, as conducted in this study, is more difficult than speaker-independent or text-independent according to the previous study [15].

A potential future research direction based on the findings in this study is to tune wav2vec 2.0 on the Japanese language with emotional contents. wav2vec is sensitive to language and could extract linguistic information within acoustic features [33]. Hence, training speech embeddings in the Japanese spoken language could be a promising research direction. This study evaluated wav2vec 2.0 pre-trained on the English emotional speech dataset (MSP-IMPROV) and then used that model to extract acoustic features from the Japanese dataset.

As for IEMOCAP, the benchmark is shown in Table 7. It shows that our results obtained in this study are competitive with the state-of-the-art results for the IEMOCAP dataset. It is worth noting that in [10], the authors combined acoustic and linguistic features that yielded higher results. Compared to recent speech embeddings like HuBERT and UniSpeech-SAT, our results without augmentation shows remarkable higher scores, highlighting the benefit of training such models on specific affective speech database. The WA and UA scores of HuBERT Large [9] were computed manually from the provided toolkit in addition to the official scores for cross-validation (CV) evaluation [6].

## 6. Conclusions

This paper reported the results of SER with different data augmentation techniques. In conclusion, adding more data does not always improve SER performance. For speaker-independent, adding more data tends to improve SER performance. For text-independent data (including spaker+text independent), the most accurate model was achieved by using only two data augmentations with glottal source filtering and start-end silence removal. For the types of data augmentations that contribute to the performance of speech emotion recognition, we found that no specific type of data augmentation performs superior to other types of data augmentations. One type of data augmentation performs better when it is combined with other types of data augmentations. We also conclude that text-independent is more difficult and challenging than speaker-independent, as previous research revealed. The result of a text-independent scenario that yields an assumption that smaller data obtains better SER performance than larger data shows the necessity to choose a proper data augmentation technique for text-independent SER.

Future research can be directed to enlarge the number of data augmentation techniques and to observe if the performance of SER can be improved with different data augmentation techniques reported in this paper. Training SER in a specific language arguably will lead to performance improvement due to the impact of the linguistic-dependency. Other augmentation techniques like SpecAugment could be explored for SER, which is not currently evaluated due to data processing differences (spectrogram vs. waveform).

## Figures and Tables

**Figure 1 sensors-22-05941-f001:**
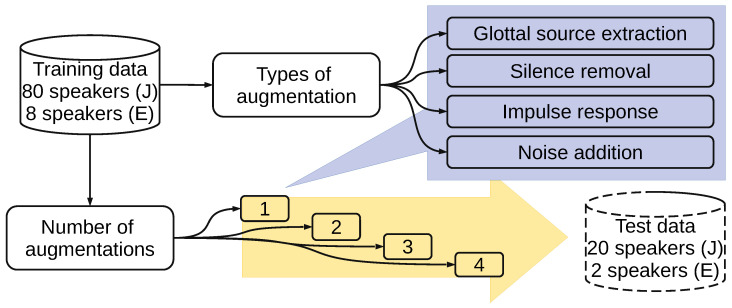
Flow of data selection for training speaker-independent SER in the experiments of data augmentation; J: Japanese (JTES); E: English (IEMOCAP); For text-independent, the split of training/test is based on sentences instead of speakers.

**Figure 2 sensors-22-05941-f002:**
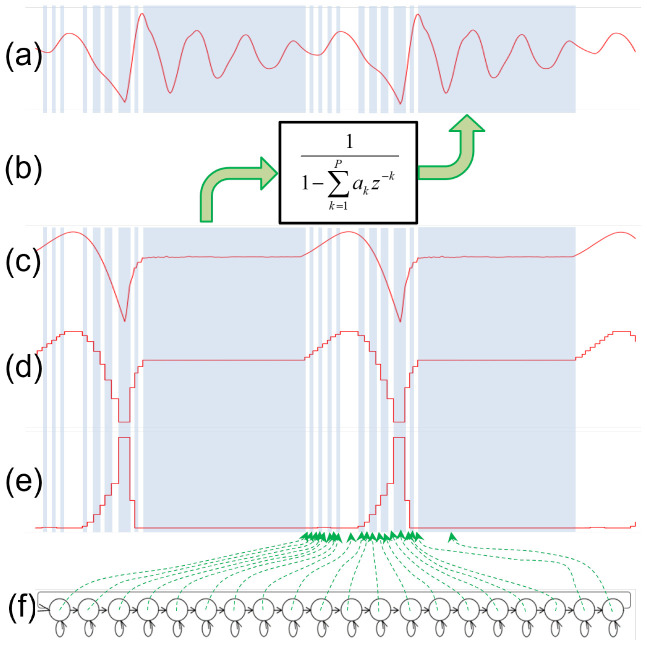
Source-filter model based on CAR-HMM. The AR filter and the HMM, as depicted in (**b**,**f**), represent a vocal tract and a generative model of a glottal flow derivative, respectively. Given a state transition sequence, the expectations and variances of each state’s output PDFs can align (for example) as depicted in (**d**,**e**). The glottal flow derivative, as depicted in (**c**), is then defined by the realized values of the aligned output PDFs. Finally, the CAR-HMM is assumed to generate the voiced speech as depicted in (**a**) by filtering the glottal flow derivative with the AR filter.

**Figure 3 sensors-22-05941-f003:**
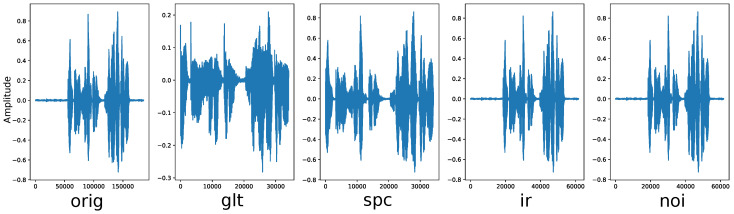
Plots of the original and augmented data from a sample in JTES dataset.

**Figure 4 sensors-22-05941-f004:**
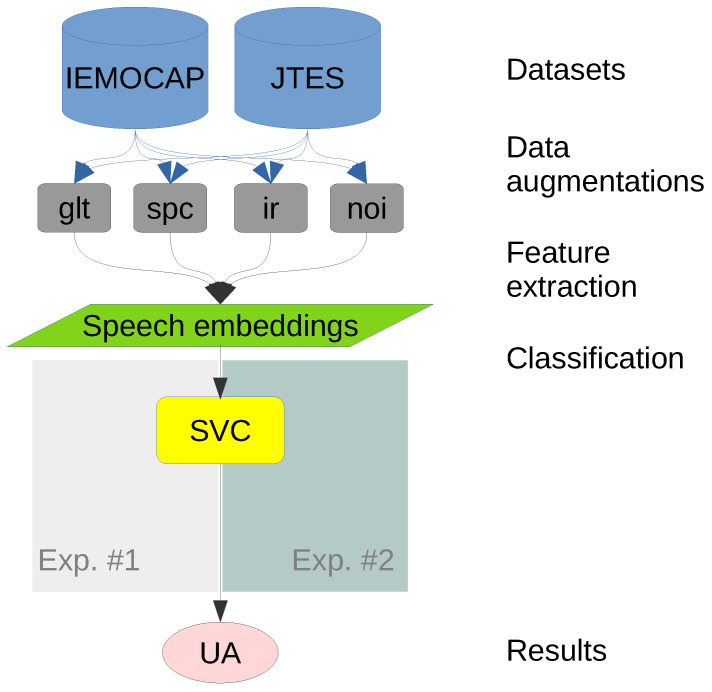
Flowchart of the main methodology from datasets to unweighted accuracy (UA).

**Table 1 sensors-22-05941-t001:** Number of data (utterances) in each number of data augmentation.

Number of Data Augmentations	Number of Training Data			
	JTES-SI	JTES-TI	JTES-STI	IEMOCAP
Without augmentation	16,000	16,000	14,400	4290
With one augmentation	32,000	32,000	28,800	8580
With two augmentations	48,000	48,000	43,200	12,870
With three augmentations	64,000	64,000	57,600	17,160
With four augmentations	80,000	80,000	72,000	21,450
Test data	2000	2000	400	1241

**Table 2 sensors-22-05941-t002:** Unweighted average recall (UAR, %) of SER with different data augmentation techniques on **JTES-SI**. orig = original JTES dataset, glt = glottal source extraction, spc = speech cleaned, ir = impulse response, noi = noise addition. The highest scores for each experiment are in bold.

Data	Exp. #1	Exp. #2
	Without augmentation	
orig	97.10	97.10
	With one augmentation	
orig + glt	96.45	96.45
orig + spc	97.20	97.20
orig + ir	97.73	97.43
orig + noi	97.55	97.60
	With two augmentations	
orig + glt + spc	96.90	96.90
orig + spc + ir	97.78	97.58
orig + ir + noi	97.85	97.83
orig + noi + glt	97.23	97.50
orig + glt + ir	97.33	97.10
orig + spc + noi	97.65	97.65
	With three augmentations	
orig + glt + spc + ir	97.53	97.40
orig + spc + ir + noi	97.98	**97.95**
orig + ir + noi + glt	97.78	97.78
orig + noi + glt + spc	97.60	97.68
	With four augmentations	
orig + glt + spc + ir + noi	**98.05**	97.90

**Table 3 sensors-22-05941-t003:** Unweighted average recall (UAR, %) of SER with different data augmentation techniques on **JTES-TI**. orig = original JTES dataset, glt = glottal source extraction, spc = speech cleaned, ir = impulse response, noi = noise addition. The highest scores for each experiment are in bold.

Data	Exp. #1	Exp. #2
	Without augmentation	
orig	75.08	75.05
	With one augmentation	
orig + glt	76.43	76.45
orig + spc	76.23	76.23
orig + ir	75.30	74.70
orig + noi	75.38	75.20
	With two augmentations	
orig + glt + spc	**76.75**	**76.75**
orig + spc + ir	75.38	74.68
orig + ir + noi	75.30	74.73
orig + noi + glt	76.05	75.98
orig + glt + ir	75.50	75.10
orig + spc + noi	75.80	75.45
	With three augmentations	
orig + glt + spc + ir	75.70	74.75
orig + spc + ir + noi	75.48	74.25
orig + ir + noi + glt	75.43	75.00
orig + noi + glt + spc	75.93	75.00
	With four augmentations	
orig + glt + spc + ir + noi	75.63	74.50

**Table 4 sensors-22-05941-t004:** Unweighted average recall (UAR, %) of SER with different data augmentation techniques on **JTES-STI**. orig = original JTES dataset, glt = glottal source extraction, spc = speech cleaned, ir = impulse response, noi = noise addition. The highest scores for each experiment are in bold.

Data	Exp. #1	Exp. #2
	Without augmentation	
orig	74.50	74.50
	With one augmentation	
orig + glt	75.50	75.50
orig + spc	76.50	76.50
orig + ir	74.75	73.75
orig + noi	76.25	75.50
	With two augmentations	
orig + glt + spc	**77.25**	**77.25**
orig + spc + ir	76.00	74.50
orig + ir + noi	75.50	74.50
orig + noi + glt	75.25	75.25
orig + glt + ir	75.00	73.75
orig + spc + noi	75.75	74.75
	With three augmentations	
orig + glt + spc + ir	76.00	74.75
orig + spc + ir + noi	74.50	74.25
orig + ir + noi + glt	74.00	75.00
orig + noi + glt + spc	75.50	75.00
	With four augmentations	
orig + glt + spc + ir + noi	74.75	74.50

**Table 5 sensors-22-05941-t005:** Unweighted average recall (UAR, %) of SER with different data augmentation techniques on **IEMOCAP**. orig = original IEMOCAP dataset, glt = glottal source extraction, spc = speech cleaned, ir = impulse response, noi = noise addition. The highest scores for each experiment are in bold.

Data	Exp. #1	Exp. #2
	Without augmentation	
orig	74.88	74.88
	With one augmentation	
orig + glt	74.23	74.23
orig + spc	75.44	75.44
orig + ir	75.03	74.80
orig + noi	75.43	75.07
	With two augmentations	
orig + glt + spc	75.66	75.66
orig + spc + ir	75.68	75.37
orig + ir + noi	75.15	75.46
orig + noi + glt	75.11	75.54
orig + glt + ir	75.01	74.5
orig + spc + noi	75.62	75.33
	With three augmentations	
orig + glt + spc + ir	75.73	75.39
orig + spc + ir + noi	76.16	**75.87**
orig + ir + noi + glt	75.16	74.54
orig + noi + glt + spc	76.03	75.71
	With four augmentations	
orig + glt + spc + ir + noi	**76.39**	75.80

**Table 6 sensors-22-05941-t006:** Comparison of JTES performances (Unweighted Accuracy, UA). The highest scores for each split are in bold.

Reference	Split	Features	Augmentation	UA (%)
[15]	SI	emo_large	No	87.88
[14]	SI	Mel-cepstrum	No	71.31
[13]	SI	ComParE	No	81.44
This study	SI	wav2vec 2.0	Yes	**97.95**
[15]	TI	emo_large	No	64.36
This study	TI	wav2vec 2.0	Yes	**76.75**
[15]	STI	emo_large	No	69.56
[16]	STI	HSFs of 5 frames	Yes	73.40
[39]	STI	LLDs	No	64.40
This study	STI	wav2vec 2.0	No	74.50
This study	STI	wav2vec 2.0	Yes	**77.25**

**Table 7 sensors-22-05941-t007:** Comparison of IEMOCAP performances. The highest scores in this study are in bold.

Reference	Test Set	Features	Aug.	WA (%)	UA (%)
[11]	CV	UniSpeech-SAT Large	No	70.78	-
[9]	CV	HuBERT large	No	67.56	-
[10]	CV	Audio_25_ + GloVe + BERT	No	77.51	78.41
[10]	Session 5	Audio_25_ + GloVe + BERT	No	83.08	83.22
[9]	Session 5	HuBERT large	No	63.90	64.54
This study	Session 5	wav2vec 2.0	No	74.13	74.88
This study	Session 5	wav2vec 2.0	Yes	**75.50**	**76.39**

## Data Availability

Not applicable.

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
