# Peer review of "Effects of Data Augmentations on Speech Emotion Recognition"

_sensors, 2022, doi:10.3390/s22165941_

Round 1
Reviewer 1 Report
Some points should be addressed to make the paper eligible for publication:
The main problem statement should be highlighted.
The author needs to improve the related work section by adding state-of-the-art related works.
The dataset visualization figure is required in the dataset section (3.1.). What are the sampling rate and other metrics of used audio files?
In section 3.2.4., what is the noise type that was added to the original dataset?
It is necessary to ad the main methodology flowchart.
What are the main differences between experiments one and two?
The results tables from 2 to 5 require more descriptions; why the results are same for some instances in both two experiments?
The ROC (Receiver Operating Characteristic) curve is required for the SVM. the author needs to calculate it.
Besides the accuracy metrics, the comparison table (benchmarking table) should contain recall, precision, and specificity. The author needs to calculate them to highlight the system quality.
Reviewer 2 Report
This paper discussed the influence of data augmentations techniques on the speech emotion recognition. Different types and numbers of data augmentation methods were tested on the Japanese Twitter-based emotional speech corpus.
The detailed comments are as follows:
1. In line 84, could the authors explain why they didn’t add the data augmentations technique in the test data?
2. In section 3.1, what’s the purpose of introducing the IEMOCAP dataset? The abstract and introduction said the paper focused on the JTES dataset.
3. Figure 3 lacks appropriate unit labels
4. In sections 3.3 and 3.4, did the JTES and IEMOCAP datasets use the same features extraction method and classifier?
5. In section 4, I think the experiment did not control the variables well, if the hardware equipment of Exp.#1 and Exp.#2 were different, then their other conditions should be controlled the same, for example the impulse response addition.
6. The effect of data augmentation presented by the experimental results does not seem to be very significant. Especially in the comparison session with previous work, the improvement in accuracy seems to come from the feature extraction method (wav2vec 2.0) rather than the effect of data augmentation.
Round 2
Reviewer 1 Report
The author completes the required modifications; the paper becomes better now. My recommendation is to accept it.
Reviewer 2 Report
The authors have addressed all of my comments. I would like to recommend it for publication in Sensors.